# Conducting Silicone-Based Polymers and Their Application

**DOI:** 10.3390/molecules26072012

**Published:** 2021-04-01

**Authors:** Jadwiga Sołoducho, Dorota Zając, Kamila Spychalska, Sylwia Baluta, Joanna Cabaj

**Affiliations:** Department of Organic and Medical Chemistry, Faculty of Chemistry, Wroclaw University of Science and Technology, Wybrzeże Wyspiańskiego 27, 50-370 Wrocław, Poland; dorota.zajac@pwr.edu.pl (D.Z.); kamila.spychalska@pwr.edu.pl (K.S.); sylwia.baluta@pwr.edu.pl (S.B.); joanna.cabaj@pwr.edu.pl (J.C.)

**Keywords:** silicon-containing polymers, silane, silole, organic light-emitting diode (OLED), organic photovoltaic (OPV), sensors

## Abstract

Over the past two decades, both fundamental and applied research in conducting polymers have grown rapidly. Conducting polymers (CPs) are unique due to their ease of synthesis, environmental stability, and simple doping/dedoping chemistry. Electrically conductive silicone polymers are the current state-of-the-art for, e.g., optoelectronic materials. The combination of inorganic elements and organic polymers leads to a highly electrically conductive composite with improved thermal stability. Silicone-based materials have a set of extremely interesting properties, i.e., very low surface energy, excellent gas and moisture permeability, good heat stability, low-temperature flexibility, and biocompatibility. The most effective parameters constructing the physical properties of CPs are conjugation length, degree of crystallinity, and intra- and inter-chain interactions. Conducting polymers, owing to their ease of synthesis, remarkable environmental stability, and high conductivity in the doped form, have remained thoroughly studied due to their varied applications in fields like biological activity, drug release systems, rechargeable batteries, and sensors. For this reason, this review provides an overview of organosilicon polymers that have been reported over the past two decades.

## 1. Introduction

Silicon, as the second most common element in the earth’s crust, forms bonds mainly with oxygen (siloxanes), hydrogen (silanes), and carbon (silanes and siloles). Due to their physicochemical properties, silicone structures have found wide application in materials science [1], optoelectronic technology [2], medicine [3], and environmental protection [4]. Compared to carbon and its bonds, e.g., with hydrogen, the Si–H bond is polarized in such way that silicon has a positive character and hydrogen has a hydride character [5]. As a result, silanes undergo heterolytic reactions more easily than alkanes. Moreover, Si–H and Si–Si bonds are weaker than their carbon analogs, while Si–X bonds (X = N, O, S, F, Cl, Br, and I) are much stronger than C–X bonds. Another difference is the instability of hydrosilicons, given the thermodynamic and kinetic results, compared to hydrocarbons [6,7]. This stability can be increased by introducing organic groups (Si–C bonds instead of Si–H). Furthermore, the Si–Si bond, due to the presence of delocalized δ-electrons, can be compared to the C = C bond and delocalization of π-electrons [8]. The siloxanes have a Si–O bond with a length of 1.64 Å, which is significantly longer than that of the C–C bond (1.53 Å). Moreover, the Si–O–Si bond angle is approximately 143°, which is much larger than the tetrahedral angle (~110°). The significant fact is that the Si–O–Si bond angle is so flexible that it can easily pass through the linear 180° state [9]. Furthermore, the torsional potential of Si–O bonds is significantly lower than that of C–C bonds. Due to these properties, siloxanes are flexible chains and have high thermal stability [10].

Recently, silanes and siloles have been used as promising building blocks for functional organic materials, since silicone-containing π-conjugated compounds are endowed with efficient electron-transporting properties and/or high quantum yields. Due to the direct relationship between the electronic structure of π-conjugated heteroaromatic systems and optoelectric properties, compounds with specific parameters can be designed. By adjusting the highest occupied molecular orbital (HOMO)/lowest unoccupied molecular orbital (LUMO) energy levels, one can control the length of the π-coupling, the frequency of light emission, as well as the injection of the charge. The silicon-containing polymers with multifunctional properties and the ease of their modification by chemical synthesis, using the most common polycondensation reactions based on Suzuki-Miyaury, Stille coupling, Heck, and Sonogashira reactions, have a wide range of potential applications in optoelectronics [11], gas separation membranes [12], porous organic frameworks [13], surface coatings [14], and sensors [15]. This review provides an overview of organosilicon polymers that have been reported over the past two decades. For a review of work in this area in the earlier years, the reader is referred to other reviews [16,17,18]. For the purposes of this review, coverage will be restricted to arylsilanes and siloles, which are increasingly used in industry and health. The work focuses on polymers that have been used mainly in organic light-emitting diode (OLED), organic photovoltaic (OPV), and sensors.

## 2. Arylsilanes—The Most Commonly Used Syntheses

Recently, a lot of attention has been paid to arylsilanes, because silicon can serve as a link between moieties in a molecule and also between organic and inorganic materials [19,20]. Such materials, due to the higher ionic character of the C–Si bond compared to the C–C bond, have greater thermal stability [21]. Moreover, molecules based on silane core have been proven to be building blocks for constructing various 3D porous metal-organic frameworks (MOFs), dynamic covalent frameworks (DCFs), element-organic framework (EOF), and dendrimer-like structures [22,23,24]. The next important group of silanes is the group of tetra-arylsilane (TPS) derivatives, due to their wide energy gap, triplet energy level, and high electrochemical, thermal, and morphological stability [25,26]. More importantly, with its higher solubility used in low-cost processing techniques, such as spin-coating and inkjet printing, desirable for large-area applications, TPS is a promising building material on an industrial scale [27,28,29].

In the synthesis of arylsilanes, palladium-catalyzed coupling reactions (e.g., Suzuki and Stille reactions) and Grignard reactions are most often used [26,30,31,32]. In order to obtain silane intermediates in the above reactions, the most common is the metalation reaction with the addition of chlorosilane additionally substituted with an aryl group or an alkyl chain [33,34,35].

This solution was used by R. Yang et al. [36]. The targeted compounds named TCzSi and DCzMzSi were synthesized by a palladium-catalyzed Suzuki–Miyaura coupling reaction (Scheme 1). The first stage was based on a metalation reaction with the addition of phenyltrichlorosilane. The results indicate that the typical tetrahedral geometry owing to the coordination of silicon was effective in ensuring the separation of the three functional units in a molecule and lead to a high triplet energy level (ET) and relatively wide band gap (Eg).

O. Bezvikonnyi and co-workers obtained 9,9′-bis(triphenylsilyl)-9H,9′H-3,3′-bicarbazole (BiCzSiPh_3_) in a two-step synthesis [37]. The first step, the synthesis of 3,3′-bicarbazole, was a Friedel–Crafts reaction with FeCl_3_ as Lewis acid. This method allows the synthesis of 3,3′-bicarbazole without using column chromatography. Compound BiCzSiPh_3_ was synthesized by standard treatment of the intermediate with n-BuLi, and then the reaction was quenched with chlorotriphenylsilane (Scheme 2).

The thermal and thermomechanical properties of the polymer containing arylsilane, bis(4-(4′-(4′-phenoxy)phenyl)phenyl)dimethylsilane phthalonitrile (SiBPPN), was studied by J. Hu et al. [38]. They obtained SiBPPN in a three-step reaction. The first stage, as previously, was a metalation reaction with the addition of dimethyldichlorosilane. The next step was the Suzuki reaction, and in the end, the SiBPPN was achieved through base-catalyzed aromatic displacement (Scheme 3).

L. H. Tagle et al. obtained poly(esters)—PEs derived from diacids containing bulky side groups, which have a halogenated (Cl, Br) imide ring, an aminoacidic residue (glycine, *L*-alanine, *L*-valine), and an amide group with a silicon-containing diphenol (Scheme 4) [39]. PEs were obtained with good yields in a typical polymerization process with pyridine, 4-toluenesulfonyl chloride, and dimethylformamide (DMF). The solubility was good in polar aprotic solvents and for some of them in tetrahydrofurane (THF). The majority showed swelling in THF and *m*-cresol.

An example of the use of the Grignard reaction is a synthesis of the silicon-containing arylacetylene resins with oxadiazole moieties [40]. In a standard procedure using magnesium powder, bromoethane, and 2,5-bis(4-(4-ethynylphenoxy)phenyl)-[1,3,4]oxadiazole, followed by an appropriate silane derivative (methylphenyldichlorosilane, dimethyldichlorosilane, and diphenyldi-chlorosilane), PODSA-MP, PODSA-MM, and PODSA-PP resins were obtained (Scheme 5).

S. A. Milenin and co-workers also used the Grignard reaction, using commercially available and cheap alkoxy and chlorosilanes, as well as *p*-bromotoluene, as the starting reagents to obtain *p*-tolylalkoxy- and *p*-tolyl(hydrido)silanes (Scheme 6) [41].

Further, based on *p*-tolylsilanes **1–6**, they obtained a series of *p*-tolylalkoxysilanes **7–12** by a variety of synthetic approaches, e.g., by a condensation of the corresponding alkoxysilanes with excess trimethylsilanol in the presence of acetic acid or by refluxing in dioxane with sodium hydroxide, followed by blocking the reaction with trimethylchlorosilane (Scheme 7). This method, as the authors claim, will find practical application in the future for synthesizing symmetric disiloxanes due to the inexpensive and commercially available reagents used and simplicity of the reaction conditions.

N. N. Makhmudiyarova et al. were the first to synthesize eight- and 11-membered cyclic silicon-containing di- and triperoxides with high yields and selectivity by the reaction of geminal bis-hydroperoxides with bis(methoxymethyl)-diphenylsilane in the presence of La(NO_3_)_3_·6H_2_O as a catalyst (Scheme 8) [42]. Eleven-membered diphenylhexaoxasiladispiro cycloalkanes were reduced to the corresponding diphenyltetraoxasiladispiroalkanes. According to the authors, cyclic peroxysilanes can also be used as hydroxylating, peroxidizing, and oxidizing agents and used for the preparation of polymers, similar to linear peroxides containing Si–O–O fragments.

## 3. Siloles—The Most Commonly Used Syntheses

Siloles possess a low-lying LUMO level, due to the σ*–π* conjugation between the π* orbital of the butadiene moiety and the σ* orbital of the two exocyclic σ-bonds on the silicon atom [43]. Furthermore, the silicon atom stabilizes the highest occupied molecular orbital; thus, siloles exhibit high thermodynamical stability in the air [44]. Moreover, aggregation-induced emission (AIE) has been observed in silole [45]. Phenyl substituted siloles and polysilanes can be considered examples of mixed π-conjugation/cross-hyperconjugation and π-conjugation/σ-conjugation, respectively.

The synthesis of dithiensiloles, as in the case of arylsiloles, is based on C–C coupling reactions, e.g., the Stille and Sonogashira reactions [46,47]. Similarly, silole-containing intermediates are synthesized by metalation in reaction with n-BuLi or reductive cyclization with lithium naphthalenide [46,48].

A. Pöcheim et al. attempt to show that the combination of siloles as the prototypical examples of cross-hyperconjugated molecules and oligosilanes, representative of σ-conjugation, forms systems of extended conjugation, leading to altered optical absorption properties [49]. Researchers decided to use di(phenylalkynyl)silane **37** as a starting material (Scheme 9). Reductive cyclization with lithium naphthalenide was followed by a reaction with excess dichlorodimethylsilane to the 2,5-chlorodimethylsilyl substituted silole **38**. If chlorotrimethylsilane was used instead of dichlorodimethylsilane, the 2,5-bis(trimethylsilyl) substituted silole **40** was obtained. The chlorodimethylsilyl substituents of **38** provide the possibility to introduce oligosilanyl groups to the 2,5-positions. A reaction with potassium tris(trimethylsilyl)silanide led to a formation of the neopentasilanyl substituted silole **39**. Compound **38** was also reacted with 1,1,4,4,4-pentakis(trime-thylsilyl)tetramethyltetrasilanylpotassium to yield compound **41** with attached oligosilanyl fragments containing linear hexasilane units.

Researchers report that the redshift of the absorption bands clearly suggests some type of conjugation between the silole and the oligosilane units. The hypothesis of conjugation between the oligosilane substituent and the silole core in compounds **39** and **41** was supported by the results of the density functional computations.

Another interesting example of silole is a work by Y. Cai et al. [50]. They report the synthesis, characterization, and photophysical properties, including aggregation-enhanced emission (AEE) characteristics of cyclotetrasiloxanes **42–44** containing different silole-based fluorogenic units (silafluorene, 1,3-diphenyl-9-silafluorene, tetraphenylsilole) (Scheme 10). 9,9-Dichloro-9-silafluorene, 9,9-dichloro-1,3-diphenyl-9-silafluorene, dichloro(methyl)phenylsilane, and 1,1-dihydroxyl-2,3,4,5-tetraphenylsilole were synthesized by cohydrolysis and condensation reactions.

Their optical properties in a solution and as crystals were studied. These compounds have low quantum yields in solution (Φ_fl_ = 0.01–0.18) with fluorescence maxima at 359–375 nm for silafluorene-containing compounds **42** and **44** and at 491 nm for AEE-active tetraphenylsilole compound **43**. However, **42–44** have high solid-state quantum yields (Φ_fl_ = 0.65–0.78) with fluorescence maxima at 377–390 nm for compounds **42** and **44** and at 517 nm for tetraphenylsilole- and silafluorene-containing compound **43**.

Recently, C. Gu et al. reported a novel conjugated polymer PDTSDTBT consisting of a donor unit with a tetrahedral Si (sp^3^) named DTS and an acceptor unit named DTBT with branched side chains [51]. The polymer was obtained by Stille coupling and then transformed into nanoparticles by nanoprecipitation (Scheme 11).

Recently, L. Yang and co-workers reported two newly synthesized chromophores, i.e., 2-(2-thienyl)-benzo[b]silole (4T1) and 2-(2,2′-bithiophen)-5-yl]-benzo[b]silole (4T2), obtained by a double trimethylsilylation of the bromobenzonitrile **46** (Scheme 12) [52]. Lithium halogen exchange, followed by treatment with trimethylborate, gave the boronic acid **47** (52% yield over three steps). The reaction of **47** with acetylene **48** (*n* = 1, 2) by rhodium(I)-catalyzed Chatani alkyne cycloaddition afforded the thienyl-appended benzosilole adducts 4T1 and 4T2 without any complication from the nitrile.

R. Cao et al. synthesized a series of functional silole derivatives via the hydrosilylation reaction of 1-methyl-2,3,4,5-tetraphenylsilole and different constructing units that contain an ethynyl group (Scheme 13) [53]. Initially, ethynyl compounds were synthesized by the Sonogashira cross-coupling reaction of the corresponding bromides, followed by the subsequent hydrosilylation reaction with 1-methyl-2,3,4,5-tetraphenylsilole at a refluxing temperature in the presence of the Karstedt’s catalyst to produce the target silole-based derivatives with satisfying reaction yields. They report that the designed compounds demonstrate two emissive wavelengths in the solution and in the solid-state, and the dual emission is tuned by the aggregation state in the mixed solvent. The compound containing silole and pyrene units shows polymorphism-dependent solid-state emissive behavior. The crystal packing of the target compounds is influenced by the steric effects of the corresponding different molecular structures.

In 2019, C. N. Scott and M. D. Bisen reported a series of 1,7-disubstituted 1,6-diynes by the Sonogashira reaction and converted them to the reactively functional 2,5-disubstituted siloles using the Kumada-type nickel-mediated intramolecular cyclization (Scheme 14) [54]. Following the preparation of the silole precursors, they pursued polymerizations of the siloles with 3,6-bis(5-bromothiophene-2-yl)-*N*,*N*-bis(2-decyltetradecyl)-1,4-dioxopyrrolo[3,4-c]pyrrole (DPP-Br) using the appropriate polymerization methods. All polymers have broad absorption spectra and reduced optical bandgaps (Eg < 2.0 eV). The desilylated polymer had a lower absorption maximum compared to the other polymers due to the absence of the silicon orbital interaction with the π-system.

## 4. Application in Different Fields

Typical optoelectronic devices are converters capable of converting the energy generated between light and electric current. Optoelectronic devices have a wide range of applications, ranging from light-emitting diodes through to photovoltaic technology, laser diodes, photoemission tubes for cameras, photoconductive cells, etc. Their extremely versatile applications require economical production on an industrial scale. Recent years have witnessed the dynamic development and enormous improvement in the efficiency of these devices and proved the enormous demand for new organic semiconductor materials, which are at the heart of such devices. In this chapter, examples of the use of silole-based compounds in optoelectronic devices will be presented.

### 4.1. Resins

The research carried out shows that the existence of the silicon-containing units decreases the melting point and improves the solubility of synthesized resin. In addition, the cured resin exhibits excellent thermal properties at temperatures as high as 546 °C under N_2_ atmosphere. The SiBPPN-containing laminate possessed a high bending strength of 389 MPa and a high interlaminar shear strength of 30.44 MPa at room temperature (Scheme 3) [38]. The resultant composite may be potentially applied as a high performance structural or functional material in aircraft, aerospace, and electronics fields.

An example of materials modified by the addition of arylsilanes are poly(esters)—PEs (Scheme 4) [39]. Despite excellent mechanical properties, chemical resistance, and thermal stability, PEs with high aromatic content are difficult to process due to high glass-transition temperature (Tg) values and insolubility in common organic solvents. The Tg and the thermal degradation temperatures values showed a tendency in the sense that when the size of the halogen atom is increased, the values decrease, due to the higher distance between the polymeric chains, which increases their flexibility and diminishes the interactions between them. PEs showed Ultraviolet/visible (UV-vis) transparence between approximately 500 and 600 nm, and those containing halogen atoms showed auto-extinction when put in the flame.

Another interesting example was presented by S. Fan and co-workers [55]. Researchers obtained an inherent flame-retardant polyamide 6 (FR-PA6) containing polydiphenylsiloxane (PDPS) under the action of ethylene glycol (EG) via a facile “two-step” bulk polymerization. Prepared FR-PA6 reached 28.3% of the limiting oxygen index (LOI) and passed the V-0 level of UL94 test with suppressed melt-dripping. After investigating the char residues and pyrolysis volatiles of FR-PA6 after combustion, the formation of a silicon-rich protective layer was proven to be an essential factor to increase the flame retardancy of FR-PA6 via reducing the diffusion of pyrolysis volatiles, restricting the heat transfer and mass loss, as well as preventing polymer melts from dripping. The novel route offered in the work is easily accessible to large-scale industrial fabrication. Polydiphenylsiloxane (PDPS) was prepared by dehydration condensation according to the literature [56], and the synthetic route is shown in Scheme 15. FR-PA6 was prepared by two-step polymerization. The chemical structure of FR-PA6 is shown in Scheme 16.

Other compounds that use the thermal properties of silanes are arylacetylene resins PODSA-MM, PODSA-MP, and PODSA-PP with 2,5-diphenyl-[1,3,4]-oxadiazole moieties, prepared by Grignard reactions in THF (Scheme 5) [40]. The resins have good processing fluidity and are soluble in common solvents such as THF, CH_2_Cl_2_, and toluene. The cured resins have good mechanical properties due to the existence of rigid-rod 2,5-diphenyl-[1,3,4]-oxadiazole and polar aromatic ether structures. The flexural strength and flexural modulus of the cured PODSA-MP resin at room temperature are 69.6 MPa and 2.5 GPa, respectively. All cured resins have good thermal and thermo-oxidative stabilities attributed to a number of benzene rings and strong charge-transfer interactions of 2,5-diphenyl-[1,3,4]-oxadiazole. Cured resins have low water uptake due to crosslinked networks. The resins possess potential application in the heat-resistant field.

### 4.2. Organic Solar Cells

Due to the increasing energy demand, solar cells have become a promising way to harness the inexhaustible clean energy of the sun. Organic solar cells (OSC) consisting of conjugated polymers or small molecules (constituting a mixture of donors) with fullerene derivatives (acting as acceptors) attract special attention due to their unique properties, such as relatively light weight, low production cost, and the possibility of producing large elastic surface. As a result, in the last decade, a significant improvement in the power conversion efficiency (PCE) above 10% was achieved [57,58,59,60,61]. In order to achieve high photovoltaic efficiency, it is necessary to maintain the specific properties of conjugated polymers; in particular, their absorption spectra and the molecular energy levels must be closely matched. The appropriate energy levels of the HOMO and the LUMO not only significantly facilitate exciton dissociation at the donor/acceptor interface, but also generate higher open-circuit voltage (V_oc_) of OSC devices [57,62]. Despite enormous advances, fullerene derivatives have some inherent disadvantages, such as poor absorption in the visible region, high synthesis cost, difficult modification of energy levels, and poor morphological stability, which makes the use of these compounds in the OSC construction problematic. Therefore, in recent years, a great deal of effort has gone into developing non-fullerene acceptors that combine good stability, easy synthesis, strong visible and near-infrared absorption, and the ability to adjust energy levels. The use of polymer donors having complementary absorption spectra with non-fullerene acceptors resulted in the PCE levels of 10–14% often reported [58,63,64]. Aromatic compounds containing silicon in their structure constitute an attractive building block due to their unique electronic structures, which consist of low LUMO energy levels and relatively small band gaps, due to the interaction between the silicon σ* orbital and the π* butadiene orbital. Because of all these properties, many conjugated polymers containing silicon in their structure were used as active substances in solar cells [57,65,66].

L. Liu et al. prepared a series of random dithienosilole-based terpolymers using two different dithienosilole monomers with a nonanoyl group and malononitrile as electron-withdrawing groups, using microwaves to aid in the polymerization of Stille coupling. Newly synthesized polymers were also tried as donor materials in polymer solar cells, all of which exhibited high open-circuit voltage above 1.0 V. The final power conversion efficiency was in the range of 1.16–3.29% with appropriately adjusted monomer proportions [57].

Dithienosilole copolymers have also been prepared by M. Li et al. A series of copolymers were designed and prepared from dithienosilole monomers with different acyl groups and benzothiophene as a comonomer (Scheme 17). A microwave-assisted Stille junction was used for polymerization. The individual polymers exhibited similar absorption behavior with slight differences. Newly synthesized polymers were assessed as donor materials in OSC, all of which showed a high open-circuit voltage of about 1.0 V. The branched modified 3,7-dimethyloctane polymer showed the best performance at a PCE of 2.66% [67].

Y. Zhao and co-workers designed and synthesized two new copolymers combining dithieno[3,2-b:2′,3′-d]silole as the donor unit and thieno[3,4-c]pyrrole-4,6-dione as the acceptor unit (Scheme 18). The group investigated the effects of various side chains on optical properties, energy levels, morphology, and photovoltaic properties. As a result, the individual polymers exhibited different levels of light and energy absorption, which had a significant effect on the short circuit current and open-circuit voltage. Finally, the maximum conversion efficiency was obtained at the level of 2.65% [64].

X. Chen et al. designed and synthesized two new small molecules of the acceptor–donor–acceptor DINDTS and DINDCNDTS containing dithienosilole as the base unit and 1,3-indanedione (IN) or a malononitrile derivative as closed-end groups (Scheme 19). These molecules were synthesized for processing in a solar cell solution with a collective heterojunction (BHJ). Optimized solar cells based on DINDTS: PC71BM were characterized by a short-circuit current of 13.5 mA cm^–2^ and an energy conversion efficiency (PCE) of 6.60% [68].

A similar donor–acceptor–donor system using 2,6-(4,4-bis-(2-ethylhexyl)-4*H*-cyclopenta[2,1-b;3,4-b′]dithiophene (DTC) and (4,4′-bis(2-ethylhexyl)dithieno[3,2-b:2′,3′-d]silole) (DTS) as a central building unit and 3-ethyl-rhodanine as closed-end groups was designed and synthesized by W. Ni and co-workers (Scheme 20). The group investigated the influence of bridging atoms on optical, electrochemical, and morphological properties. Differences of single atoms in the structure of polymers resulted in changes in individual properties. An optimized solar cell based on DR3TDTS showed a PCE above 8% [69].

Examples of the use of polymers containing silicon in their structure for the construction of organic semiconductors (OSCs) are presented in Table 1.

The above studies confirm that single changes in the polymer structure (different side chains or core groups) can have a huge impact on the properties of the obtained polymers in terms of their application in OSCs. The use of different alkyl side chains in the synthesis of polymers means that the obtained compounds exhibit different levels of light and energy absorption, which affects the short-circuit current and open-circuit voltage, respectively. The introduction of the alkyl chain into the π-bridge causes a significant increase in the dihedral angle between the mote and the donor unit, which may ultimately affect the value of the maximum energy conversion efficiency. The parameters of individual polymers are varied, which proves that the selection of appropriate side chains is of great importance when designing high-performance OSCs. Similar properties are exhibited by polymers with different groups in the core. Despite slight differences in the structure of the central core units (e.g., the silicon atom in the DTS block and the carbon atom in the DTC block [69]), individual polymers showed very different properties, including membrane absorption, molecular packing, and charge transport. The silicon atom showed much better absorption and packing properties than the carbon atom.

### 4.3. Organic Light-Emitting Diodes

Over the last three decades, organic light-emitting diodes (OLEDs) have enjoyed great interest in the industry due to their high potential for use in flat panel displays and semiconductor lighting [71,72].

OLED is a type of semiconductor lighting system that converts electric current into light. The emitting layer in semiconductors used in OLEDs is based on fluorescent, phosphorescent materials or TADF (thermally activated delayed fluorescence) [73].

OLEDs can be divided into two basic groups. The first group consists of small molecule-based OLEDs (MOLEDs), while the second group is polymer-based OLEDs (PLEDs). It should be mentioned that organic light-emitting materials significantly affect the efficiency of devices. Equipment performance is strongly dependent on the inherent physicochemical properties of molecular aggregates. The efficiency of emission and the ability to carry the medium in solid films are particularly important [71,74,75]. When comparing molecules or polymers with ACQ (aggregation-caused quenching) properties, active AIE materials show numerous advantages, such as high external quantum efficiency (EQE), relatively easy synthesis, unadulterated production, and simplified device configuration [75].

Among all materials belonging to AIE, silole-based compounds are readily used in OLEDs due to their high thermal stability, good electronic mobility, and affinity, and moreover, they exhibit high solid-state fluorescence quantum efficiency—some compounds achieve efficiency up to 100% [76,77].

R. Yang et al. synthesized tetraphenylsilane derivatives (TCzSi and DCzMzSi) (Scheme 1) [36]. A TCzSi-based blue phosphorescent organic light-emitting diode (PHOLED) exhibited a maximum current efficiency (CE) of 25.5 cd A^−1^, maximum power efficiency (PE) of 18.15 lm W^−1^, maximum luminance of 6285 cd m^−2^, and maximum external quantum efficiency of 13.32%. These results reveal that highly twisted tetraarylsilane compounds have great potential for highly efficient PHOLEDs as host materials.

On the other hand, O. Bezvikonnyi and co-workers synthesized two new 3,3′-bicarbazole derivatives functionalized with triphenylsilyl and with tosyl moieties attached to 9,9‘positions of 3,3′-bicarbazole moiety, and they measured their thermal, optical, photophysical, electrochemical, and charge-transporting properties (Scheme 2) [37]. Electron mobility of 5.6 × 10^−4^ cm^2^/(Vs) was observed for the layer of BiCzSiPh_3_ at the electric field of 5.4 × 10^5^ V/cm, while hole mobility was recorded to be 1.4 × 10^−4^ cm^2^/(Vs) at the same electric field. BiCzSiPh_3_-based phosphorescent OLED exhibited power and external quantum efficiencies as high as 25 lm/W and 13.8%, respectively.

H. Park et al. synthesized a group of dithiensiloles based on fluorene and carbazoles (DTS), which were then tested for their thermal, photophysical, and electrochemical properties (Scheme 21). All properties confirmed that the compounds could be used as basic materials for red emitters in OLED. All compounds showed reversible oxidation and had low localized LUMO energy due to the conjugated fluorene/carbazole substituents in DTS. This, together with high fluorescent quantum efficiency, confirms the possibility of using these compounds in OLEDs as emitters, carriers, and materials for electron transport [78].

Another series of carbazole-substituted dithienosiloles was synthesized by Y. Xiong et al. (Scheme 22). A thorough analysis of the crystal and electronic structure, thermal stability, and electrochemical and photophysical properties confirmed the significant properties of AIE with high emission efficiency from solid layers, thanks to which these compounds can be used in the design of light-emitting layers in undoped OLEDs (electroluminescence efficiency at the level of 17.59 cd/A, 12.55 lm/W, and 5.63%) [79].

2,3,4,5-tetraarylsiloles are a class of very important luminogenic materials characterized by efficient solid-state emission and excellent electron transportability. B. Chen et al. introduced the bulky 9,9-dimethylfluorenyl, 9,9-diphenylfluorenyl, and 9,9′-spirobifluorenyl substituents at the 2,5 positions of the silole rings (Scheme 23). Compounds modified in this way showed higher fluorescence quantum yields (2.5–5.4%) than 2,3,4,5-tetraarylsiloles. The group also constructed efficient OLEDs using synthesized compounds as main emitters, which ensured high luminance, current efficiency, and energy efficiency at the level of 44,100 cdm^−2^, 18.3 cdA^−1^, and 15.7 lmW^−1^, respectively [80].

H. Nie together with the group designed and synthesized four silole derivatives: (PBI) 2DMTPS, (PBI) 2MPPS, (PPI) 2MTPS, and (PPI) 2MPPS (Scheme 24), which are characterized by high AIE activity and emission in solid films, and also show good thermal stability and low energy LUMO. Three-layer OLED without doping (PPI) 2DMTPS has excellent electroluminescence (EL) parameters with an efficiency of 15.06 cd A^−1^, 16.24 lm W^−1^, and 4.84%. Importantly, similar parameters to the three-layer OLED (13.30 cd A^−1^, 14.51 lm W^−1^, and 4.25%) were obtained by an efficient two-layer OLED based on (PBI) 2DMTPS. The obtained results confirmed the effectiveness of the designed compounds in using them as n-type light emitters for efficient, simple, and inexpensive OLEDs [71].

M. Ju et al. compared three dimethyltetraphenylsiloles with symmetrically substituted acceptor or donor moieties (Scheme 25). Of all the compounds tested, the multi-layer organic light emitting diode using DMTPS-DPA showed the highest efficiency, luminance, intensity, and internal efficiency: 3.1 V, 13,405 cd m^−2^, 8.28 cd A^−1^, 7.88 lm W^−1^, and 2.42%, respectively. DMTPS-DPA can also be used in hole transporting layers due to the high mobility of holes [81].

Examples of the use of polymers containing silicon in their structure for the construction of OLED are presented in Table 2.

Studies of individual electrochemical, thermal, and photophysical properties of newly synthesized compounds based on DTS have shown that the introduction of conjugated fluorene/carbazole substituents into the systems has an impact on obtaining high thermal stability, intense green emission, and reversible electrochemical behavior. Moreover, the differences in the connections between the silole and carbazole groups have little effect on the changes in optical properties, making it possible to obtain efficient OLEDs using different combinations of silole and carbazole derivatives.

### 4.4. Organic Field-Effect Transistors

Work on improving and tuning the performance of organic field-effect transistors (OFETs) has attracted a lot of attention over the last decade. The key elements in constructing OFET are low production costs, long service life, and high mobility of the load. Compared to the small molecules used to construct OFET by vacuum deposition, high molecular weight conjugated polymers are a promising alternative for the cheap and convenient production of OFET. Hence, conjugated polymers have become promising candidates for OFET materials [83,84,85].

The necessary condition for the construction of a long-life OFET device is the use of small particles or polymers characterized by high thermal, photo-air, and morphological stability [83].

To date, many types of OFET based on polymer semiconductors have been developed and described. Due to the type of main charge carriers, polymer semiconductors can be divided into p- and n-type semiconductors and bipolar semiconductors, capable of transporting holes, electrons, and both, respectively. When selecting the organic semiconductor used in OFET construction, the HOMO and the LUMO energy levels of the organic semiconductor molecule in relation to the work function of the contact electrode material are important. So far, the techniques of producing p-type materials have been the most popular; however, more and more attention has been paid to modern techniques of n-type materials synthesis [86,87,88].

H. Zhang et al. designed two conjugated donor–acceptor π polymers with 4,4′-bis(2-ethylhexyl)-5,5′-bis(trimethyltin)dithieno[3,2-b:2′,3′-d]silole as donor and isodiketopyrrolo-pyrrole or isodithioketopyrrol-pyrrole as acceptor moiety (Scheme 26). Newly synthesized polymers were examined for their optical and electrochemical properties and their potential use in the construction of OFET. The work showed that isodithioketopyrrolopyrroles are promising building blocks for efficient organic field-effect transistors (hole up to 0.49 cm^2^V^−1^s^−1^ and electron mobility up to 0.26 cm^2^V^−1^s^1^) [83].

K. Kim et al. used the Stille coupling reaction to synthesize two copolymers consisting of diketopyrrolopyrrole (DPP) and silole derivatives (Scheme 27). The newly synthesized polymers were tested for their electrical properties in OFET and circuits. After spinning, both OFETs showed quite low values of hole mobility, while in the case of their annealing at a temperature of 150 °C, typical properties of bipolar transport were observed with average values of hole and electron mobility of 1 × 10^−1^ and 2 × 10^−3^ cm^2^/(Vs). The above parameters confirm that the synthesized compounds can be used in OFET [86].

Y. C. Pao et al. designed and synthesized a tricyclic diseleno[3,2-b:2′3′-d]silole (DSS) in which the 3,3′position is bridged by dioctylsilole moieties (Scheme 28). The DSS modified in this way was copolymerized with a diketopyrrolopyrrole acceptor (DPP), as a result of which a donor–acceptor copolymer was obtained. Selenophene based PDSSDPP shows the mobility of 2.47 × 10^−2^ cm^2^V^−1^s^−1^, suggesting that the DSS unit and its polymers are promising for use in OFET [89].

Examples of the use of polymers containing silicon in their structure for the construction of OFET are presented in Table 3.

The above examples confirm that a small, single difference of atoms or functional group in the structure of polymers can cause drastic changes in optical and electronic properties. In the example reported by H. Zhang et al., thionation of the product not only significantly improved the mobility of the charge, but also made the polymer bipolar transport properties [83]. Moreover, quantum chemical calculations and electrochemical analysis have confirmed that thionation has an additional effect on optimizing molecular boundary orbitals and facilitating charge introduction.

### 4.5. Silicone-Based Materials in Sensors and Biosensors

Biosensors are devices that help to assess the levels of biological markers or any chemical reaction by producing the signals that are mainly related to the concentration of an analyte in the chemical reaction. This kind of device is usually used to monitor diseases and treatment progress, and to detect disease-causing bacteria and markers, based on the measurement from body fluids (e.g., saliva, blood, and urine) [91]. The receptor layer of each biosensor is built from a biorecognition material (e.g., enzyme or DNA), which recognizes the desired analyte, whereas the transducer converts the (bio)chemical signal resulting from the interaction of the analyte with the receptor into a digital electronic signal (Figure 1) [92].

The key step in the biosensors design is the proper process of immobilization of the biological material on a suitable matrix, which should not only effectively bind the biomolecule on its surface, but also prevent the weakening of its catalytic activity and ensure stability during measurements as long as possible. The manipulation of nanostructure materials in conjunction with biological molecules has led to the development of a new class of hybrid modified sensors, in which both enhancement of charge transport and biological activity preservation may be preserved. Conductive polymer’s charge transfer capacity acts as an excellent matrix for biomolecules, providing an enzyme mimetic environment [93]. Due to the presence of aromatic units in the polymer backbone, the immobilization is performed with the help of π–π stacking interactions of the polymer and enzymatic protein. These strong interactions effectively stabilize the tertiary structure of proteins [94].

Conjugated heterocycles are often used in the fabrication of layered biosensors as electronic mediators to improve the contact between the active center of the enzyme and the electrode surface. The silane-based compounds represent new promising building blocks for modifying sensing systems [95]. Dithienosilole derivatives, compared to many building units of π-conjugated polymers, such as furan, pyrrole, or pyridine, show the lowest level of LUMO energy and a relatively high level of HOMO orbitals, which means that compounds based on dithienosilole form stable films on solid substrates and they improve the charge transfer (CT) [46,96]. CT-type reactions, thanks to electron holes (h) and electrons (e^−^), are the basic element of the work of biosensor devices, using enzymes from the oxidoreductase class, catalyzing reactions based on the transport of the hopping charge, and/or tunnelling of charge carriers [97]. Depending on the type of redox reaction being catalyzed, charge transport can take place to (reduction) or from (oxidation) the enzyme. According to the theory, electron holes and electrons are transported through the HOMO and LUMO orbitals.

Tetraphenylsilane derivatives have a tetrahedral geometry—they are built from phenyl rings connected to a silicon atom in the sp^3^ configuration, which decides the possibility of their use in the construction of three-dimensional, porous organic materials [24,98]. Tetraphenylsilane as the core of the monomer unit with high triplet energy and wide band gap represents a group of charge-transferable compounds [99,100,101]. In this context, the silicon atom present in the structure of organosilicon compounds breaks the coupling between various building blocks so that the donor molecular orbitals do not overlap with the acceptor molecular orbitals (charge separation), while ensuring targeted charge transport between the donor and acceptor moieties [102,103].

The C–Si bond (σ bond) in tetraphenylsilane is parallel to the π bonding orbitals; therefore, an interaction between σ–π bonds is possible. In addition, tetraphenylsilane has a high energy of the first excited state, up to 3.0 eV, resulting from an interruption of conjugation in the multiple bond system by the silicon atom. This leads to the inhibition of electron interactions between the unit made from tetraphenylsilane and neighboring groups [30]. The width of the energy gap (Eg = 4.2 eV) and the possibility of attaching a coupled silyl–aryl core and electron–donor and electron–acceptor systems to σ–π allow obtaining a material with ambipolar properties, which can be easily modified [104]. Until now, tertaphenylsilane derivatives have been mainly studied for their use in light-emitting diodes, but the above-mentioned optoelectronic and physicochemical properties make them an excellent candidate for use in medicine and biosensors. Y. Shang et al. reported that the star-shaped amphiphilic poly[2-(diethylamino)ethylmethacrylate]-b-poly[2-hydroxyethyl methacrylate]s copolymers, using tetraphenylsilane as a core, exhibit merits in comparison with the linear counterpart, including enhanced micelle stability, high drug-loading, and entrapment efficiency [105]. The π–π interaction between tetraphenylsilane cores results in excellent stability of micelles assembled from the star-shaped copolymers with critical micelle concentration (CMC) values in the range from 1.49 to 3.93 mg L^−1^. The in vitro experiments demonstrate that the drug-loaded micelles possess pH-response function and can effectively release a drug in a simulated cancer cell environment. The blank copolymers micelles are nearly non-cytotoxic, while the drug-loaded micelles have significant anti-cancer effects.

Polymers based on tetraphenylsilane and dithiensiloles structures show the ability of oxidative polymerization, creating stable films on solid substrates [34,46]. Introduction of aryl groups, including thiophene, to organosilicon derivatives can improve the stability of the films based on these compounds, which is extremely important from the point of view of the construction of receptor matrices in biosensors. The studies performed earlier showed that applying tetraphenylsilane in an optical bio-sensing system, as well as dithienosilole derivatives in electrochemical biosensors, significantly improved working parameters of designed bio-devices. The microfluidic optical biosensor described by S. Baluta et al. [106] was based on the fluorescence sensing strategy for dopamine (DA) detection. The idea of measurement was based on forming the polydopamine (poly(DA)) film on the surface of graphene quantum dots (GQDs). Prepared GQDs were highly luminescent due to the aromatic planar structure. The ceramic-based miniature biosensor was designed and constructed through the immobilization of laccase on an electrochemically synthesized polymer-poly(bis(4-(thiophen-2-yl)phenyl)diphenylsilane), based on low temperature co-fired ceramics technology (LTCC), prepared as presented by K. Malecha [107]. This sensing system utilized the catalytic oxidation of DA to dopamine-o-quinone (DOQ), and then to poly(DA) (in alkaline conditions), which can selectively quench the strong luminescence of GQDs due to Förster Resonance Energy Transfer (FRET). The detection limit of the biosensor constructed this way was equal to 80 nM, and the system showed a broad linear range (1–200 µM) with insignificant influence of interference species on the detection (Figure 2A,B).

Other results presented by S. Baluta et al. [108] concern the electrochemical enzymatic platforms based on dithienosilole derivatives (Scheme 29), which allow monitoring of serotonin (5-HT) and DA. Electrochemical determination of serotonin and dopamine was achieved using cyclic voltammetry (CV) (to present a whole redox process, applied potential in range −0.2–0.8 V) and differential pulse voltammetry (DPV) (used for linear range determination, applied potential in range −0.2–0.8 V) techniques.

A platinum electrode modified with poly(2,6-bis(3,4-ethylenedioxythiophen-5-yl)-4-methyl-4-octyl-dithienosilole) (Scheme 29A) film and laccase was used to selectively determine serotonin. The dithienosilole derivative had been polymerized onto platinum electrode electrochemically (via CV in the potential range 0–1.4 V). Laccase was immobilized onto the modified electrode with a thin film of semiconducting polymer. This investigation confirmed that serotonin undergoes a catalytic redox reaction by laccase. This was crucial, as the electrochemical oxidation of serotonin by laccase has never been described in detail in the scientific literature. The electrochemical nature of 5-HT was examined in a wide range of concentrations (0.1–200 μM), employing the CV method (applied potential −0.2–0.8 V, scan rate = 50 mV/s) in oxygen-saturated conditions. The bio-platform prepared this way presented good working parameters, such as working in a wide range of concentrations (0.1–200 μM) and showing a very good detection limit of −48 nM.

The second bio-platform, a gold electrode modified with a thin poly(2,6-bis(selenophen-2-yl)-4-methyl-4-octyl-dithienosilole) (Scheme 29B) film and horseradish peroxidase (HRP)), made for dopamine detection, also presented a very good detection limit of −73 nM in a wide concentration range (0.1–200 μM). Both systems presented high selectivity.

Investigations of obtained polymer films based on silicone-based structures onto solids, performed with AFM and SEM techniques, showed no visible surface defects and significantly regular surface of both polymers. The diameter of the resulting grains of both polymers was in a range of 1.5–8 μm. These diameters allow enzyme molecules to anchor freely into the polymeric films while maintaining their catalytic activity and permit the rotation of the direct active site of the enzyme towards the substrate. Provided investigations showed that silole derivatives can be used as a matrix for anchoring the biological elements and can be applied in biosensors.

The large diversity of functional groups that can be attached or obtained in self-assembled monolayers (SAMs) represent a new means of surface chemistry preparation. Currently, a high interest in SAMs is observed due to their ability to alter the surface properties. They are supramolecular aggregates that can create ultrathin organic film on the solids; the most investigated combinations include thiols on gold and alkylsilanes on silicon [109,110,111].

X. Zhong et al. described a novel glucose biosensor designed with self-assembling of a double-layer 2D-network of (3-mercaptopropyl)-trimethoxysilane (MPS), gold nanoparticles, and glucose oxidase (GOD), which was studied on a gold substrate [112]. The researchers prepared a bare gold electrode to obtain a self-assembled monolayer in the first step and a 2D-network by immersing it in a special solution containing MPS dissolved in ethanol and by adding NaOH in the second step. As the final step, the gold nanoparticles were chemisorbed onto the thiol groups present in the second silane layer. In the end, the enzyme-GOD was physically immobilized on the surface of the gold nanoparticles. This prepared bio-platform was used for glucose detection with electrochemical impedance spectroscopy (EIS) and CV techniques. The results showed that the constructed biosensor system exhibits a good stability and sensitivity and can detect glucose in a wide concentration range of 0.0004–0.528 µM.

Silane-based compounds are not only used as matrices for the production of bio-platforms for biosensor applications. C. Hoffman and G. Tovar presented a different approach to the research of this group of compounds [113]. They focused on non-specific adsorption of proteins to the surface for better understanding of the exact quantity and activity of a protein—the control over the adsorption behavior of proteins in contact with solid surfaces. Researchers described monolayers from the newly synthesized compounds based on methoxy-tri(ethyleneglycol)-undecenyldimethylchlorosilane and dodecyldimethylchlorosilane (DDMS) (Scheme 30A,B), both mixed. Obtained SAMs were investigated for the possibility of enhanced protein repellent properties, e.g., Ras Binding Domain (RBD) was used, a protein with high relevance for cancer diagnostics. This protein was physically adsorbed onto the silicon oxide surfaces silanized with DDMS or non-silanized silicon wafers; however, no RBD protein was linked to surfaces silanized with created compounds (methoxy-tri(ethyleneglycol)-undecenyldimethylchlorosilane and dodecyldimethylchlorosilane (DDMS)).

Generally, the organosilanes start to be widely used in biosensors as hetero-bifunctional cross-linkers. Mostly, these compounds are composed of two distinct reactive moieties: the silyl head group and the organic reactive group classically carried at the end of an aliphatic chain (tail group). The head group containing Si can react with the solid surface (-OH groups) and create a strong attachment of the organosilane molecule to the surface. The second group (tail chain), which is an organic functional group, e.g., carboxyl-, amino-, and epoxy-, allows the biologically active material’s immobilization, which allows the biomolecular recognition of the desired analyte [114].

S. Corrie et al. confirmed the advantage of organosilane bifunctionality by reported spectroscopic study involving the chemical and structural modification of thiol-functionalized organosilica particles with aminosilane to produce a bifunctional silica hybrid [115]. They presented a multiplexed model bioassay that shows bifunctionality, enabling separate covalent attachment strategies for both homogeneous incorporation of fluorescent dyes and surface-specific biomolecule attachment.

Another study, performed by C. Corso et al., was based on using two different silane molecules, (3-glycidyloxypropyl)trimethoxysilane (GPS) and (3-mercaptopropyl)trimethoxysilane (MTS), for the immobilization of fluorescently labeled IgG antibodies onto planar ZnO surfaces [116]. According to the obtained results, the immobilization of antibodies onto a film based on MTS gave 21% higher fluorescence response in comparison to GPS. Due to this, an MTS-based bio-platform was successfully used in the acoustic biosensor tests.

The results presented by S. Baluta et al., described in this section, show that biosensors constructed with silicone-based materials represent very promising building blocks in the case of creating a proper bio-platform for monitoring and detecting various biologically active species. This group of compounds strongly improve working parameters of biosensors, such as detection limit, selectivity, sensitivity, and lasting, in comparison with other detection bio-systems, where a wide range of different conducting compounds, including nanomaterials, were used [117,118,119,120]. Obtaining better parameters is strongly linked with the structure of silicone-based materials. In the case of electrochemical biosensors, the matrix based on conducting derivatives is expected to facilitate electron transfer, thereby enhancing the sensor sensitivity. Charge carrier transfer reactions are vital in biosensors. The redox-active proteins are based on the transfer of charge carriers by hopping and/or long-range tunneling [97]. Due to the potential for oxidation or reduction processes that are suitable in a biosensor, both types of CT can be transferred to/from the enzyme. Due to this, presented compounds improve an electron transfer in the case of electrochemical biosensors, which is the basis of working of such bio-tools. Silicone-based derivatives represent an excellent matrix for application in biosensors because they can act, due to the structure, as semiconductors that can transfer electrons from oxidized species, which are detected by the long-range direct tunneling mechanism.

## 5. Conclusions

Organosilicon polymers are a group of high-molecular inorganic–organic compounds of high technological importance. This group of compounds with multifunctional properties has a wide range of potential applications. Organic materials deposited on a silane core are characterized by good mechanical and thermal properties. These compounds have a number of applications, including microelectronics, organic optoelectronics, biomedicine, surface coatings, and sensors. The article systematically reviews the synthesis methods and the important physical properties of silanes and siloles that have been developed over the last two decades. The most important synthetic methods in this field are based on C–C coupling and metalation reactions. Silicone-containing polymers create stable films on solid substrates and improve charge transfer. In addition, the sp^3^ silicon atom present in the structure of the silane compounds breaks the coupling between the different building blocks, while ensuring targeted charge transport between the donor and acceptor moieties. Product thioning (DTS derivatives [83]) not only significantly improved the mobility of the charge, but also made the polymer a bipolar transport property. Moreover, the introduction of the alkyl chain into the π bridge causes a significant increase in the dihedral angle between them and the donor unit, which may ultimately affect the value of the maximum energy conversion efficiency in the OPV. The parameters of individual polymers are varied, which proves that the selection of appropriate side chains is of great importance when designing high-performance OSCs. It should be emphasized that silicon-containing materials are building blocks that can be produced on an industrial scale. In addition, thanks to the use of a wide range of polymers containing silanes with different properties in the design, biosensors can be used in diagnostics, for the detection of single molecules as well as a large group of compounds. Such systems (e.g., dithienosilole and tetraphenylsilane) act as an ideal matrix for anchoring biological material, because they not only effectively bind to its structure, but also improve the performance parameters of bio-recognition elements. Due to good mechanical, thermal, and conductive properties, as well as application properties, systems based on silanes arouse more and more interest of scientific teams.

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
