# Peer review of "Conducting Silicone-Based Polymers and Their Application"

_molecules, 2021, doi:10.3390/molecules26072012_

Round 1
Reviewer 1 Report
This review article is dedicated to silicone polymers.
In a first part, the authors describe the various arylsilanes and siloles synthesized in the literature.
Then, in a second part, they describe the applications of these compounds in organic solar cells, organic light-emitting diodes and organic field-effect transistors.
This review is well structured, clear, contains a lot of information and fully describes the literature.
Only the conclusion (which is really short) needs to be improved, in particular by proposing perspectives and trends for the future.
Author Response
The Reviewers of Molecules
Dear Reviewer,
Due to the submitting our article to Molecules (09.03.2021) in special issue: “Conducting Polymers”, titled Conducting silicone-based polymers and their application – Jadwiga Sołoducho, Dorota Zając, Kamila Spychalska, Sylwia Baluta, Joanna Cabaj and received details of the revisions (15.03.2021) we would like to present our responses to the Reviewers' comments.
First of all, we would like to thank for the detailed review, moreover, we would like to inform, that all significant changes in the manuscript have been highlighted. Also, the English language has been checked and improved.
According to comments of Reviewer 1 we would like to introduce our responses:
Point 1: This review article is dedicated to silicone polymers. In a first part, the authors describe the various arylsilanes and siloles synthesized in the literature. Then, in a second part, they describe the applications of these compounds in organic solar cells, organic light-emitting diodes and organic field-effect transistors. This review is well structured, clear, contains a lot of information and fully describes the literature. Only the conclusion (which is really short) needs to be improved, in particular by proposing perspectives and trends for the future.
Response 1: We are truly sorry for all the mistakes. We would also thank you for your valuable comments. The conclusion was improved (the changes in text were highlighted). We place the changed text below:
Organosilicon polymers are a group of high-molecular inorganic-organic compounds of high technological importance. This group of compounds with multifunctional properties has a wide range of potential applications. Organic materials deposited on a silane core are characterized by good mechanical and thermal properties. These compounds have a number of applications, including microelectronics, organic optoelectronics, biomedicine, surface coatings, and sensors. The article systematically reviews the synthesis methods and the important physical properties of silanes and siloles that have been developed over the last two decades. The most important synthetic methods in this field are based on C-C coupling and metalation reactions. Silicone-containing polymers create stable films on solid substrates and improve charge transfer. In addition, the sp3 silicon atom present in the structure of the silane compounds breaks the coupling between the different building blocks, while ensuring targeted charge transport between the donor and acceptor moieties. Product thioning (DTS derivatives [84]) not only significantly improved the mobility of the charge but also made the polymer a bipolar transport property. Moreover, the introduction of the alkyl chain into the π bridge causes a significant increase in the dihedral angle be-tween them and the donor unit, which may ultimately affect the value of the maximum energy conversion efficiency in the OPV. The parameters of individual polymers are varied, which proves that the selection of appropriate side chains is of great importance when designing high-performance OSCs. It should be emphasized that silicon-containing materials are building blocks that can be produced on an industrial scale. In addition, thanks to the use of biosensors in the design – a wide range of polymers containing silanes with different properties, such devices can be used in diagnostics, for the detection of single molecules as well as a large group of compounds. Such systems (e.g. dithienosilole, tetraphenylsilane) act as an ideal matrix for anchoring biological material, because they not only effectively bind to its structure, but also improve the performance parameters of bio-recognition elements. Due to good mechanical, thermal, and conductive properties, as well as application properties, systems based on silanes arouse more and more interest of scientific teams.
We hope, that this response will satisfy the Reviewer 1. In addition, we would like to thank for all the valuable comments.
Finally, we would like to apologize for all the errors and omissions that have appeared in the manuscript and thank you for the valuable reviews. We hope that the answers given to the detailed comments of the Reviewers will be satisfactory. Once again, we would like to emphasize that all changes have been included in the manuscript (highlighted) and the manuscript has been corrected again in terms of language.
Looking forward to hearing from you soon.
Sincerely yours,
Jadwiga Sołoducho
e-mail address: jadwiga.soloducho@pwr.edu.pl
Reviewer 2 Report
The manuscript by Soloducho and colleagues concerns the applications of a class of polymers, i.e. conducting silicon polymers, that worth the publication in this journal (even more in this specific special issue). On the other hand, In order the manuscript to be considered further, it needs to be improved. The manuscript should be a review and not just a list of papers on the topic, with a short coverage of any of them.
Apart on the need of a general English language editing, these are points that should be addressed to make the manuscript more attractive and better organized:
- with respect to the first section, an introduction to the topic "conducting" is completely missing. In the present way, the introduction only refers to an encyclopedia-type definition of si-containing polymers. Please try to be more specific, as expected for a review of a specific topic. It is not even clear what the aim of the review is: try to justify it, otherwise it just look like a list of papers.
-in the sections 2 and 3 the authors review the synthesis of arylsilanes and silols, respectively, speaking about their applications in the following section, in some cases also reporting the synthesis of some specific polymer. Well, that is confusing for the reader … I would suggest to introduce the specific section devoted to the structures giving some general, and short, overview of the methodology of synthesis focused on the structure/properties that may be obtained. Then moving to the applications sections, where in some cases the synthesis may be specifically detailed.
- More in general, in a manuscript like this one, comments/critical evaluation of the available literature is expected. Particularly important papers/knowledge must be critically presented/highlighted with personal comments by the authors otherwise it just remains a bibliographic research as those made by degree/master degree students. A review must be an help, with critical information (not a list) on a specific topic.
In addition, the conclusion section of a review is probably the most important must contains comments, again emphasizing the most important results obtained so-far, giving an outlook, perspectives, personal vision (by the authors), etc. Of course the content of this section must be directly related with the aim introduced in the first one.
Author Response
The Reviewers of Molecules
Dear Reviewer,
Due to the submitting our article to Molecules (09.03.2021) in special issue: “Conducting Polymers”, titled Conducting silicone-based polymers and their application – Jadwiga Sołoducho, Dorota Zając, Kamila Spychalska, Sylwia Baluta, Joanna Cabaj and received details of the revisions (15.03.2021) we would like to present our responses to the Reviewers' comments.
First of all, we would like to thank for the detailed review, moreover, we would like to inform, that all significant changes in the manuscript have been highlighted. Also, the English language has been checked and improved.
Due to the comments obtained from Reviewer 2, we are also presenting our answers:
Point 1: The manuscript by Soloducho and colleagues concerns the applications of a class of polymers, i.e. conducting silicon polymers, that worth the publication in this journal (even more in this specific special issue). On the other hand, In order the manuscript to be considered further, it needs to be improved. The manuscript should be a review and not just a list of papers on the topic, with a short coverage of any of them.
Apart on the need of a general English language editing, these are points that should be addressed to make the manuscript more attractive and better organized:
- with respect to the first section, an introduction to the topic "conducting" is completely missing. In the present way, the introduction only refers to an encyclopedia-type definition of si-containing polymers. Please try to be more specific, as expected for a review of a specific topic. It is not even clear what the aim of the review is: try to justify it, otherwise it just look like a list of papers.
Response 1: Thank you very much for your thorough text analysis and valuable comments. We are truly sorry for all the mistakes. In addition, we would like to inform, that the English language has been checked and improved.
In addition, the introduction was improved (the changes in text were highlighted, line 45). Recently, silanes and siloles are currently used as a promising building block for functional organic materials since silicone‐containing π‐conjugated compounds are endowed with efficient electron‐transporting properties and/or high quantum yields. Due to the direct relationship between the electronic structure of π-conjugated heteroaromatic systems and optoelectric properties, we can design compounds with specific parameters. By adjusting the highest occupied molecular orbital (HOMO) / lowest unoccupied molecular orbital (LUMO) energy levels, you can control the length of the π-coupling, the frequency of light emission as well as the injection of the charge. The silicon-containing polymers with multifunctional properties and the ease of their modification by chemical synthesis using the most common polycondensation reactions based on Suzuki-Miyaury, Stille coupling, Heck, and Sonogashira, have a wide range of potential applications in optoelectronics [11], gas separation membranes [12], porous organic frameworks [13], surface coatings [14], and sensors [15].
Point 2: In the sections 2 and 3 the authors review the synthesis of arylsilanes and siloles, respectively, speaking about their applications in the following section, in some cases also reporting the synthesis of some specific polymer. Well, that is confusing for the reader … I would suggest to introduce the specific section devoted to the structures giving some general, and short, overview of the methodology of synthesis focused on the structure/properties that may be obtained. Then moving to the applications sections, where in some cases the synthesis may be specifically detailed.
Response 2: Thank you very much for this valuable notice.
The sections 2 and 3 have been rearranged in line with the comments of the reviewer (the changes in text were highlighted, line 76 to 296).
Point 3: More in general, in a manuscript like this one, comments/critical evaluation of the available literature is expected. Particularly important papers/knowledge must be critically presented/highlighted with personal comments by the authors otherwise it just remains a bibliographic research as those made by degree/master degree students. A review must be an help, with critical information (not a list) on a specific topic.
Response 3: The comments were placed at the end of the subsections in section 4, in line with the comments of the reviewer (the changes in text were highlighted, line: 370, 471, 530, 717).
line 370: The above studies confirm that single changes in the polymer structure (different side chains or core groups) can have a huge impact on the properties of the obtained polymers in terms of their application in OSCs. The use of different alkyl side chains in the synthesis of polymers means that the obtained compounds exhibit different levels of light and energy absorption, which affects the short-circuit current and open-circuit voltage, respectively. The introduction of the alkyl chain into the π-bridge causes a significant increase in the dihedral angle between the mote and the donor unit, which may ultimately affect the value of the maximum energy conversion efficiency. The parameters of individual polymers are varied, which proves that the selection of appropriate side chains is of great importance when designing high-performance OSCs. Similar properties are exhibited by polymers with different groups in the core. Despite slight differences in the structure of the central core units (e.g. the silicon atom in the DTS block and the carbon atom in the DTC block [70]), individual polymers showed very different properties, including membrane absorption, molecular packing and charge transport. The silicon atom showed much better absorption and packing properties than the carbon atom.
line 471: Studies of individual electrochemical, thermal and photophysical properties of newly synthesized compounds based on DTS have shown that the introduction of conjugated fluorene / carbazole substituents into the systems has an impact on obtaining high thermal stability, intense green emission and reversible electrochemical behavior. Moreover, the differences in the connections between the silole and carbazole groups have little effect on the changes in optical properties, making it possible to obtain efficient OLEDs using different combinations of silole and carbazole derivatives.
line 530: The above examples confirm that a small, single difference of atoms or functional group in the structure of polymers can cause drastic changes in optical and electronic properties. In the example reported by Zhang et al. thionation of the product not only significantly improved the mobility of the charge but also made the polymer bipolar transport properties [84]. Moreover, quantum chemical calculations and electrochemical analysis have confirmed that thionation has an additional effect on optimizing molecular boundary orbitals and facilitating charge introduction.
line 717: The results presented by Baluta et al., described in this section show that biosensors constructed with silicone-based materials represent very promising building blocks in the case of creating a proper bio-platform for monitoring and detecting various biological active species. This group of compounds strongly improve working parameters of biosensors, such as detection limit, selectivity, sensitivity, lasting, in comparison with other detection bio-systems, where were used a wide range of different conducting compounds, including nanomaterials [120-123]. Obtaining better parameters is strongly linked with the structure of silicone-based materials. In the case of electrochemical biosensors, the matrix based on conducting derivatives is expected to facilitate electron transfer, thereby enhancing the sensor sensitivity. Charge carrier transfer reactions are vital in biosensors. The redox-active proteins are based on the transfer of charge carriers by hopping and/or long-range tunneling [98]. Due to the potential for oxidation or reduction processes that are suitable in a biosensor, both types of CT can be transferred to/from the enzyme. Due to this, presented compounds improve an electron transfer in the case of electrochemical biosensors, which is the basis of working such bio-tools. Silicone-based derivatives represent an excellent matrix for application in biosensors because they can act, due to the structure, as semiconductors which can transfer electrons from oxidized species, which are detected, by the long-range direct tunneling mechanism.
Point 4: In addition, the conclusion section of a review is probably the most important must contains comments, again emphasizing the most important results obtained so-far, giving an outlook, perspectives, personal vision (by the authors), etc. Of course the content of this section must be directly related with the aim introduced in the first one.
Response 4: The conclusions were improved (the changes in text were highlighted, line 736). We place the changed text below:
Organosilicon polymers are a group of high-molecular inorganic-organic compounds of high technological importance. This group of compounds with multifunctional properties has a wide range of potential applications. Organic materials deposited on a silane core are characterized by good mechanical and thermal properties. These compounds have a number of applications, including microelectronics, organic optoelectronics, biomedicine, surface coatings, and sensors. The article systematically reviews the synthesis methods and the important physical properties of silanes and siloles that have been developed over the last two decades. The most important synthetic methods in this field are based on C-C coupling and metalation reactions. Silicone-containing polymers create stable films on solid substrates and improve charge transfer. In addition, the sp3 silicon atom present in the structure of the silane compounds breaks the coupling between the different building blocks, while ensuring targeted charge transport between the donor and acceptor moieties. Product thioning (DTS derivatives [84]) not only significantly improved the mobility of the charge but also made the polymer a bipolar transport property. Moreover, the introduction of the alkyl chain into the π bridge causes a significant increase in the dihedral angle be-tween them and the donor unit, which may ultimately affect the value of the maximum energy conversion efficiency in the OPV. The parameters of individual polymers are varied, which proves that the selection of appropriate side chains is of great importance when designing high-performance OSCs. It should be emphasized that silicon-containing materials are building blocks that can be produced on an industrial scale. In addition, thanks to the use of biosensors in the design – a wide range of polymers containing silanes with different properties, such devices can be used in diagnostics, for the detection of single molecules as well as a large group of compounds. Such systems (e.g. dithienosilole, tetraphenylsilane) act as an ideal matrix for anchoring biological material, because they not only effectively bind to its structure, but also improve the performance parameters of bio-recognition elements. Due to good mechanical, thermal, and conductive properties, as well as application properties, systems based on silanes arouse more and more interest of scientific teams.
We hope, that all our responses will be suitable for the Reviewer’s 2 comments. In addition, we would like to thank for the Reviewer’s valuable review.
Finally, we would like to apologize for all the errors and omissions that have appeared in the manuscript and thank you for the valuable reviews. We hope that the answers given to the detailed comments of the Reviewers will be satisfactory. Once again, we would like to emphasize that all changes have been included in the manuscript (highlighted) and the manuscript has been corrected again in terms of language.
Looking forward to hearing from you soon.
Sincerely yours,
Jadwiga Sołoducho
e-mail address: jadwiga.soloducho@pwr.edu.pl
Round 2
Reviewer 2 Report
All the suggested changes/critical comments have been addressed by the authors